# Joint Acquisition Estimator of Modern GNSS Tiered Signals Using Block Pre-Correlation Processing of Secondary Code

**DOI:** 10.3390/s20102965

**Published:** 2020-05-23

**Authors:** Jiří Svatoň, František Vejražka

**Affiliations:** CTU in Prague, FEE, Department of Radioengineering, Technicka 2, 160 00 Prague, Czech Republic; vejrazka@fel.cvut.cz

**Keywords:** block signal processing, GNSS signal acquisition, parallel code search algorithm, pre-correlation coherent accumulation, secondary code, single block zero-padding

## Abstract

Objective is a joint primary and secondary code (SC) acquisition estimator of tiered Global Navigation Satellite Systems (GNSS) signals. The estimator is based on the Parallel Code Search algorithm (PCS) combined with the Single-Block-Zero-Padding (SBZP) and the Pre-correlation Coherent Accumulation (PCA). The PCA realizes the extension of the coherent integration time in front of the PCS. However, the PCS with the SBZP and the PCA is affected by a navigation/SC bit transition problem due to its cyclic property of a computed Cross-Ambiguity Function (CAF). This CAF is degraded by diverse parasitic fragments and is not directly applicable for an acquisition. A novel analysis of this mechanism and its impact is presented. Then, the proposed modified SBZP (mSBZP) modified PCA (mPCA) PCS estimator is constructed, which does not degrade the CAF. The mSBZP allows the use of the PCS algorithm in the presence of SC bit transition, while the mPCA decreases the number of PCS algorithm calculations by a factor of SC chip count due to SC pre-correlation processing. The algorithm has the same detection performance in comparison with conventional Double-Block-Zero-Padding (DBZP). However, it allows using the PCS of half-length with longer latency up to a factor of SC chip count.

## 1. Introduction

The Global Navigation Satellite Systems (GNSS) is a primary source of position information in many different technical and scientific applications [1]. Reception is required even in many challenging and weak signal environments [2]. These applications request a new class of signals, which are suitable for an extension of the coherent integration time (Pre-detection Integration Time, PIT) to improve receiver sensitivity. So-called GNSS tiered signals are used for these purposes.

Tiered signals [3,4] use a primary code that is period-by-period modulated by bits of a periodic secondary code (SC). The SC has correlation properties like the primary one, but it is usually considerably shorter. The SC is used for data-less pilot channels (e.g., Galileo E1C) mainly, but its utilization in data channels is currently possible as well (e.g., BeiDou B1-I). Examples of SC are in Table 1. Autocorrelation functions of selected SC are depicted in Figure 1. The utilization of SC has the following essential benefits [4]: Pilot signals can be coherently correlated over its period without navigation message bit transitions. Therefore, SC pilot signals can increase receiver sensitivity;Tiered signals have better cross-correlation properties in comparison with other GNSS signals;A tiered signal spectrum has better immunity to narrow-band jamming. The power spectral density is shaped by the sinc function but is line-like due to the primary code periodicity. These lines are more “diluted” by an additional period of the SC code.The SC improves and speeds up the bit-synchronization. Some tiered signals are also equipped by a navigation message (Table 1). The full correlation of the SC results in automatic navigation message bits synchronization [4].

The most critical part of receiver signal processing is signal acquisition and its sensitivity [2]. Tiered signal acquisition is based on an extension of PIT over SC bits. It allows us to increase acquisition sensitivity, but it also poses other problems. One of them is that a PIT extension increases requirements for a search in many Doppler frequency shift bins.

Commonly used approaches (for the tiered signal acquisition) are based on a two-step schema. The primary code is acquired by the help of a non-coherent averaging in the first step [4,5]. Then, the SC is acquired through the extension of PIT in a post-correlation stage in the second step. It can be realized in a sequential or a parallel PCS-way as well [6]. This approach does not increase PIT. Therefore, the number of Doppler frequency shift bins does not rise too. However, the primarily intended benefit, an improvement of acquisition sensitivity, is not achieved here because of a loss caused by utilization of a non-coherent averaging in the first step. 

Single-step based approaches search in primary and SC phase domains concurrently. Such a joint estimator utilizes SC bits to increase PIT. The traditional sequential search algorithms [7,8] are time-consuming to search in both domains. Thus, modern approaches try to adopt any parallelization.

The PCS algorithm based computation of CAF is parallelized for all code phases using the Fast-Fourier-Transform (FFT) algorithm [9,10,11,12] and different types of zero-padding methods [2]. The signal is processed over blocks of the primary code period. However, the not-known position of SC bit transition within the processed block is a problem. It causes a rise of parasitic fragments in the CAF, which decreases the detection performance. This effect is a consequence of the cyclic property of a PCS-computed correlation.

The goal is the single-step PCS-computed joint estimator of both primary and SC phases, which suppresses the SC bit transition problem of the PCS algorithm. The secondary motivation is a realization of the extension of PIT in a pre-correlation, a pre- PCS stage. The benefit is a considerable reduction of the number of PCS algorithm calculations by a factor of a SC chip count. However, it leads again to a sequential search in the SC phase. Fortunately, SC is significantly shorter than the primary one. 

The novel approach of this article is the explanation and a principle of suppression of negative effects of a SC bit transition effect to the PCS, and the extension of PIT in the pre-PCS processing of SC for practical utilization.

## 2. State of the Art 

This article focuses on methods based on a block pre-correlation accumulation of a signal by the PCA together with the PCS algorithm. This type of a block pre-correlation processing is based on the principle that has been presented before under names:Block averaging pre-processing [13];Pre-correlation (coherent) accumulation (PCA) [14];Block-accumulating coherent integration over extended interval (BACIX) [15].

The same principle under these different names was used for a weak signal acquisition [11,16], but its utilization for tiered signals has never been considered. The PCA is an alternative way for a realization of a PIT extension in a pre-correlation stage. It is realized using a coherent accumulation of samples over blocks of primary code period of signal period-by-period together. Then, FFT blocks work just once per a SC period. The advantages are energy and implementation savings. FFT blocks can then be realized with weaker latency requirements [17]. 

The modified version (mPCA) on signals with SC was used at first in [17,18]. It accumulates primary code periods coherently with respect to estimated SC bits. The SC is then searched sequentially.

Utilization of the PCS algorithm and zero-padding together with the presence of SC or a navigation data bit transition has a consequence in the CAF (also called the Generalized cross-correlation acquisition function) [19,20,21,22,23]. However, these effects, together with a pre-correlation SC processing, have not been thoroughly studied before. Thus, it was considered in this article. 

Contributions [17,24] present both types of an estimator schema, the two-step, and the single-step based. Both schemas were analyzed together for both types of zero-padding methods (SBZP and DBZP) and SC processing in both pre- and post-correlation stages. Similar approaches based on a realization in the Field Programmable Gate Array (FPGA) were also presented in an article [18]. However, this approach was based only on DBZP. The intended pre-correlation processing single-step joint primary and SC phase estimator was not presented yet.

## 3. Common SC Acquisition Methods Theory

The acquisition is a rough estimation (^) of the primary code phase (*τ*), the Doppler carrier frequency shift (*f_d_*), and the SC phase (*n*) set of received signal parameters. The *n* could be expressed as an integer part of the code phase in integer periods *T* of the primary code phase *τ*. This expression is beneficial according to the chosen processing of a tiered signal over blocks of primary code periods.

The estimation (1) is realized using the CAF function *R*[*τ*,*f_d_*,*n*] (2) between the received signal *r*[*t*] and its locally generated replica. The replica consists of the periodic primary code *c*[*t*] with the period of *T* samples, and the periodic *SC*[*t*] with the period of *N* bits. Apriori unknown navigation message could be present in some types of signals (Table 1). The symbol (⊕) denotes the modulo-2 (xor) product. The star (*) is a symbol of complex conjugation. The “arg max” is the argument of the maxima operator. The “mod” is the modulo operator. The *f_d_* is usually searched sequentially, and it is not the subject of this article.
(1)$$\left[\hat{\tau },{\hat{f}}_{d},\hat{n}\right]=\arg\underset{\tau ,f_{d},n}{\max}\left\{R\left[\tau ,f_{d},n\right]\right\}$$
(2)$$\begin{array}{l} R\left[\tau ,f_{d},n\right]={\sum_{t=0}^{\left(N-1\right).T}r\left[t\right]e^{-j2\pi f_{d}t}SC\left[t-nT\right]⊕c\left[t-\tau \right]}^{*}\\ \mathrm{where}:n=\mathrm{integer}\left(\frac{\tau }{T}\right)\mathrm{mod}N \end{array}$$

The sequential search algorithm in Figure 2 is a straightforward realization of (2) in the time domain. This approach is time-consuming. However, the calculated CAF is linear-like and has no losses.

### 3.1. Acquisition—PCS Algorithm, Zero-Padding Algorithms, and Bit Transition Problem

The PCS algorithm works with the help of the Wiener-Khinchin-Einstein-Kolmogorov theorem (3). The Discrete Fourier transform (DFT) is used to compute the power spectral density *C_x_* from the discrete signal autocorrelation function *R_x_*. The Equation (3) could be rewritten to a form (4). This form uses the FFT instead of the DFT for a speedup. The symbol (°) denotes the Hadamard product.
(3)$$C_{x}\left[f\right]=\mathrm{DFT}\left\{R_{x}\left[\tau \right]\right\}=\sum_{\tau =0}^{T-1}R_{x}\left[\tau \right]\;e^{-\frac{j2\pi }{T}f\tau }$$
(4)$$R\left[\tau ,f_{d}\right]=\mathrm{IFFT}\left\{\mathrm{FFT}\left\{r[t]\right\}\circ \mathrm{FFT}{\left\{c[t]e^{j2\pi f_{d}t}\right\}}^{*}\right\}$$

The FFT and its inverse operation (IFFT) is realized as the RADIX-2 or -4 algorithm for an optimal implementation. However, these algorithms require the number of samples, the *N_FFT_*, to be equal to 2*^a^*, where *a* is an integer. Nevertheless, it is usually not equal to the number of samples in the primary code period *T*. The solution is the zero-padding [25] of the *r*[*t*] and *c*[*t*] blocks of signal samples with *N_z_* zeroes. 

The SBZP uses a single period of the primary code (5). It preserves a good resolution of a computed cross-correlation function. The resolution is the number of samples of a PCS-computed cross-correlation function per one primary code chip. However, this type of zero-padding causes a formation of false parasitic peak fragment in a code phase domain of the resulting CAF. This fragment is a copy of the original main peak and is at a pitch equal to the number of padded zeroes *N_z_*. An amplitude of the false parasitic peak is at the expense of the main peak. The loss is proportional to the *τ* [14,17,20]. The false peak has an amplitude higher than the main peak for *τ* > (1/2)*T*. This false parasitic fragment peak has an origin in a violated cyclic property of a signal, which is caused by the added zeroes. A similar negative consequence has the navigation message or SC bit transition in the Doppler shift domain.

The DBZP uses two periods of a signal (6), and at least one is always correlated despite the bit transition. Thus, DBZP has no extra losses. However, the algorithm requires a double-sized *N_FFT_* where half of its result is discarded anyway. A DBZP utilization for acquisition of signals with a potential bit transition is popular. New optimizations of the algorithm exist [18,21]. The DBZP works as a single-step post-correlation SC estimator [21,23]. It is suitable for further comparisons. However, its combination with the PCA and usability for SC acquisition is limited and problematic [17]. It is discussed in Section 4.
(5)$$\begin{array}{l} SBZP:\\ r[t]=\left[\begin{array}{cc} r[1xT] & 0\left[1xN_{FFT}-T\right] \end{array}\right]\\ c[t]=\left[\begin{array}{cc} c[1xT] & 0\left[1xN_{FFT}-T\right] \end{array}\right] \end{array}$$
(6)$$\begin{array}{l} DBZP:\\ r[t]=\left[\begin{array}{ccc} r[1xT] & r[1xT] & 0\left[1x2N_{FFT}-2T\right] \end{array}\right]\\ c[t]=\left[\begin{array}{ccc} c[1xT] & 0\left[1xT\right] & 0\left[1x2N_{FFT}-2T\right] \end{array}\right] \end{array}$$

The original (4) could be rewritten to (7). It is a straightforward single-step post-correlation approach. The g(*t*,*i*) is an auxiliary window function which consists of time-shifted Heaviside step functions *H*[*t*]. It is a mathematical model of a selection of *i*-th consecutive blocks of the received signal *r*[*t*] of length *T* for a correlation. The schema in Figure 3 realizes (7). The symbol “conj” denotes complex conjugation.
(7)$$\begin{array}{l} R\left(\tau ,f_{d},n\right)=\sum_{i=0}^{N-1}SC[\left(i-n\right)\mathrm{mod}N]\mathrm{IFFT}\left\{\mathrm{FFT}\left\{r[t]g\left(t,i\right)e^{-j2\pi f_{d}t}\right\}\circ \mathrm{FFT}{\left\{c[t]\right\}}^{*}\right\}\\ \mathrm{where}:g\left(t,i\right)=H\left[t-iT\right]-H\left[t-\left(i+1\right)T\right] \end{array}$$

The presence of the bit transition causes a splitting of the main and the false parasitic peak of the SBZP (in the code phase domain) even in the Doppler shift domain of the CAF. The original sinc(2π*f_d_NT*) shaped lobe in the Doppler shift domain is split into two side-lobes with a pitch equal to 2/(*NT*) Hz and a lower amplitude, as in Figure 4. These effects in the CAF are problematic for the utilization of the PCS as the joint estimator. This phenomenon in the CAF is well described in [19], using the rectangular (REC) *W*-function (8). 

The *W*-function models the effect of bit transition. It is the non-return-to-zero (NRZ) REC function. The same SC bit sign of *r*[*t*] and *SC*[*t*] in time is represented by its positive value, and different SC sign by negative value (8). The splitting of the CAF in the Doppler shift domain is equal to the DFT of the *W*-function (9). The bit transition affects the CAF in the Doppler shift domain so much that the *f_d_* estimation is biased. This function is illustrated in Figure 4, together with the corresponding CAF for *τ* = (1/2)*T*. The “*E*” means normalized CAF cross-correlation energy. The study of SC bit transition problem on a model using the *W*-function is a key factor for its effective suppression.
(8)W[t,τ]= SC[t−τ] SC[t]
(9)$$\mathrm{DFT}\left\{W\left[t,\tau \right]\right\}=\sum_{t=0}^{T-1}W\left[t,\tau \right]e^{-j\frac{2\pi }{T}f_{d}t}$$

The effects mentioned above were illustrated using simulations of the straightforward single-step post-correlation approach schema (Figure 3). Two metrics for an algorithm comparison were chosen; the Peak-to-Noise-Ratio (*PNR*) (10) and the First-to-Second-Peak-Ratio (*FSPR*) (11). These metrics express ratios in the CAF. Both metrics are suitable for signal detection and a comparison of algorithms behavior [26]. The “max,” “mean,” and “max_second_” are operators for maximum, mean, and second maximum peak value of the CAF.
(10)$$PNR=\frac{\max\left\{\left|R\left(\tau ,f_{d}\right)\right|\right\}}{\mathrm{mean}\left\{\left|R\left(\tau ,f_{d}\right)\right|\right\}}$$
(11)$$FSPR=\frac{\max\left\{\left|R\left(\tau ,f_{d}\right)\right|\right\}}{\max_{\mathrm{second}}\left\{\left|R\left(\tau ,f_{d}\right)\right|\right\}}\;$$

### 3.2. Acquisition—Common SC PCS Post-Correlation Methods—Results

The results of the straightforward single-step post-correlation approach using the schema in Figure 3 are presented in Figure 5 for both the SBZP and the DBZP algorithms. The detailed view of the first two SC bits is provided in Figure 6. The *N_FFT_* equal to the 16K (2^14^) FFT, *N_z_* = 384, and base-band Galileo E1C signals with CS25_1_ and 4 MHz sampling rate were used for simulation in this article.

The inappropriate behavior of the SBZP algorithm, the dependency of its peak amplitude on *τ*, is evident. The DBZP provides a better result. Its *PNR* value is constant across a proper SC bit (*τ* = <0,*T*>). Therefore, the DBZP is potentially more suitable and can work as the post-correlation joint primary and SC phase estimator. However, practical utilization is limited for its worse performance in the *FSPR*.

Previous schemas are the post-correlation SC processing based. Nevertheless, this article focuses on effective pre-correlation processing of SC using the PCA principle.

## 4. Pre-Correlation SC Processing, PCA, mPCA

All considerations about the processing of SC and the extension of PIT in a pre-correlation, a pre-PCS stage, are based on the principle of linearity. Then, the FFT (IFFT) and the sum operation could be commuted, and Equation (7) could be rewritten to the form shown in (12).
(12)$$R\left(\tau ,f_{d},n\right)=\mathrm{IFFT}\left\{\mathrm{FFT}\left\{\sum_{i=0}^{N-1}SC[\left(i-n\right)\mathrm{mod}N]r[t]g\left(t,i\right)e^{-j2\pi f_{d}t}\right\}\circ \mathrm{FFT}{\left\{c[t]\right\}}^{*}\right\}$$

### 4.1. PCA and mPCA

The PCA (*y*[*t*]) is a period-by-period coherent accumulation of *N* blocks–periods of primary code of length *T* of the signal *r*[*t*] (13). Such a system with the impulse response (14) is linear, but it is not time-invariant. The δ[*t*] is the Kronecker delta function. The symbol *Z* denotes integer numbers.
(13)$$y[t]=\sum_{i=0}^{\left(N-1\right)T}r[t-iT]$$
(14)$$h[t]=\sum_{i=0}^{\left(N-1\right)T}\delta \left[t-iT\right]=\left\{\begin{array}{c} 1,\left(\frac{t}{T}\right)\in Z\\ 0,otherwise \end{array}\right\}$$

The PCA unit, which is modified to be controlled by a SC bit sign, is called the mPCA. It was presented in [17]. The manner of a period-by-period accumulation that respects SC bits is depicted in Figure 7. Such the SC processing is a type of the pre-correlation processing.

A realization in hardware is beneficial, since it requires only one extra Multiplier-ACcumulator (MAC) block, as is shown in Figure 8. The MAC is an expensive hardware resource of FPGA or any other digital signal processing system. In this case, the MAC unit with an expensive complex multiplier could be minimized to the adder and the Random-Access-Memory (RAM). This advantageous hardware simplification is allowed by a binary property of used GNSS codes.

The PCA principle is simple and well known. However, it is applicable only for block-oriented signal processing over blocks of signal samples stored in a memory. It is also the case of the PCS algorithm based acquisition unit. The previous post-correlation acquisition schema from Figure 3 (7) could be redrawn to the pre-correlation schema in Figure 9 (12), which utilizes the mPCA unit from Figure 8. The difference is only in the accumulator position.

### 4.2. PCS Algorithm Using mPCA Pre-Correlation Processing of SC—Results

Results using the schema in Figure 9 with the mPCA are presented in Figure 10 for the application of the SBZP and the DBZP.

Results using the mPCA and the SBZP were equivalent to the post-correlation schema results in Figure 5, respectively Figure 6. It is a consequence of the principle of linearity. These results are still a function of the *τ*. Hence, the SBZP is still unsuitable as the intended joint estimator. However, the mPCA DBZP results in Figure 10 are different from the previous post-correlation results from Figure 5 and Figure 6. This phenomenon conflicts with the principle of linearity. The inappropriate DBZP behavior is illustrated by its CAF for *τ* equal to (1/4)*T* and (3/4) of *T* in Figure 11. 

The straightforward solution combining the PCS and the mPCA failed for both the SBZP and the DBZP algorithms. These results were already presented in [17]. Thus, the two-step approach with PCS and SBZP was used. The SC bit transition problem in the second step was eliminated by synchronization of a primary code period with the *τ*, which was acquired in the first step. Therefore, the rest of this article focuses on a SBZP utilization as a mentioned single-step estimator. The aim of the following section is to provide an analysis of the adverse effects that complicate its exploitation.

## 5. Analysis of the Adverse Effects on the PCS-SBZP Algorithm CAF

The ideal CAF is shaped as one sharp peak in the code phase domain, and the sinc(2π*f_d_NT*) function shaped lobe in the frequency-Doppler shift domain. The *PNR* and the *FSPR* ratios are given only by the cross-correlation function of the used primary code and the sinc function given by a PIT time *NT*. However, SBZP results presented above differed. Their CAFs contained parasitic fragments that decreased the *PNR* and the *FSPR*. An analysis of bit transition effects exists [19], but the overall analysis of the effect in periodic SC has not been published yet.

The effects which were introduced in Section 3.1 are called spreading or splitting [19,22] of the original main CAF cross-correlation peak. The parasitic fragment, false peak in the code phase domain, is a general effect of the SBZP algorithm. The original main correlation peak in the code phase domain is divided into the main and the false peak (Section 3.1). The side-lobes in the Doppler shift domain are caused by the SC bit transition and could be modeled by the following REC functions, by their spectra respective. The resulting CAF is a combination of these spectra and the general effect of the SBZP.

The behavior of the *W*-function (Section 3.1) over SC bits (Figure 12) models the splitting of the parasitic false peak in the CAF also along the Doppler shift domain. Its spectrum is illustrated in Figure 13 (left) for a case of the Galileo CS25_1_ code and *τ* = (1/4)*T*. The side-lobes are repeated with a lower amplitude on many frequencies. These frequencies are given by a quasi-periodicity of the SC. It decreases the *PNR* and the *FSPR* performance, side-lobes cause the biased *f_d_* estimation again. 

A new *W*’-function represents the cross-correlated part of the signal and its replica. The *W*’-function (15) is the return-to-zero (RZ) REC function and is illustrated in Figure 12. Its spectrum is given by (16). Its behavior affects the main peak and its loss. The sum of *W*’-functions over *NT* is periodic with *T*. Therefore, the CAF of the main peak in the Doppler shift domain is formed by the original sinc(2π*f_d_NT*) lobes that are situated on multiples of 1/*T* frequency and are weighted by sinc(2π*f_d_N*(*T*−*τ*)) (16). The splitting of the main peak in the Doppler shift domain of the CAF is illustrated for *τ* = (1/4)*T* in Figure 13 (right). The CAF contains spectral fragments with amplitude depending on *τ*. Such spectral fragments decrease the *FSPR* too.
(15)W’[t,τ]=H[t−iT]−H[t−iT+τ]
(16)$$\begin{array}{l} \mathrm{DFT}\left\{\sum_{t=0}^{\left(N-1\right)T}W’\left[t,\tau \right]\right\}=\;\sum_{t=0}^{\left(N-1\right)T}W’\left[t,\tau \right]e^{-\frac{i2\pi }{T}f_{d}t}\\ \approx \mathrm{sinc}\left[\pi f_{d}\left(T-\tau \right)\right]\delta \left[f_{d}\pm n\frac{1}{T}\right],\tau \in [0,T) \end{array}$$

The overall CAF in the Doppler shift domain is modeled by both the *W* and the *W*’ spectra. The main peak is split by the spectrum of *W*’-function, and the false peak by the spectrum of *W*-function. This hypothesis is supported by the following 3D CAF and its cuts in both the code phase and the Doppler frequency shift domains.

Figure 14 is an illustration of the CAF for *τ* = (1/4)*T*. The effect of the *W*’-function on the dominant main peak is evident. It is represented by parasitic spectral fragments of the sinc(2π*f_d_NT*) lobe on multiplies of 1/*T* frequency in the Doppler shift domain. The small false peak in the code phase domain is contained too.

Figure 15 is the CAF for a *τ* = (3/4)*T* and illustrates the dominant value false peak in the code phase domain. It is split in the Doppler shift domain by the spectrum of *W*-function with side-lobes. The main peak with a very low amplitude is included too. However, the amplitude of the false peak with side-lobes is higher. Therefore, the estimator of parameters *τ*, *f_d_* is biased.

The presented results were computed for *N_FFT_* = 16 384, *N_z_* = 384, and Galileo E1C signal with CS25_1_. However, the results in the Doppler shift domain were almost equal even for the case *N_z_*= 0. In this case, the false peak in the code phase domain was not present. It occurred because the false parasitic peak in the code phase domain is a product of a non-zero *N_z_*. Nevertheless, the effects of *W* and *W*’-functions are a product of the SC bit transition, and therefore a splitting remained.

The amplitude of the resulted CAF main peak is equal to the mean value of the *W*’-function (the sum). Then, this *W*’-function mean value can be used to express the loss (17) in decibels as a relative decrease in the peak amplitude. It also corresponds with the loss in the CAF’s main peak in Figure 14 and Figure 15. The loss 2.5 dB is for *τ* = (1/4)*T* in Figure 14, and 12 dB for *τ* = (3/4)*T* in Figure 15.
(17)$$Loss\left[\tau \right]=-20\log_{10}\left(1-\frac{1}{T}\sum_{t=0}^{T-1}W’\left[t,\tau \right]\right)=-20\log_{10}\left(1-\frac{1}{T}\sum_{t=0}^{\tau -1}1\right)=-20\log_{10}(1-\frac{\tau }{T});\;\tau \in [0,T)$$

The straightforward PCS—PCA approach fails due to the effects of the *W* and the *W*’-function to the CAF. The proposed solution algorithm, which suppresses these adverse effects, is presented in the next section.

## 6. Proposed PCS Algorithm with mSBZP and mPCA

The approach trying to overcome the failure of methods above was presented in [17] and [24]. However, details and derivations of the CAF were not presented. Study [17] used the PCA for SC processing and identified problems caused by SC bit transition at first. However, the approach did not find an appropriate single-step solution. The approach in [24] tried to apply a partial overlap of a resulting cross-correlation function for a replica and a replica with a shifted SC. Nevertheless, the approach was not successful, and the derivation was not presented.

### 6.1. Modified SBZP Joint Primary and SC Estimator (mSBZP for SC Algorithm)—Theory

The *W*-function models a formation of the false peak fragment. The approach of suppression of its effect is based on differences in phase-spectra of two *W*-functions (*W*_1_, *W*_2_) for the original and its time-shifted version. Their power spectra could be the same, but phase-spectra differ. Let the *W*_2_-function be complementary to the original *W*_1_-function in the time domain. Then, the coherent sum of the *W*_1_-function and the *W*_2_-function spectra is equal to zero. This is illustrated in Figure 16 using *W*-functions in the time and the spectral domain. Therefore, the parasitic effect on the CAF is subtracted, and the false peak is not formed and split. A similar effect is observable when two cross-correlation functions for two consecutive SC bits are combined.

The proposed algorithm is called mSBZP for SC and works as follows. At first, the cross-correlation function over SC period *NT* is computed for an estimated *n*. Then, the second cross-correlation function is calculated for the same received signal, but using the consecutive SC phase *n* + 1 in the replica. Then, its result is circularly shifted back by *N_z_* samples. At last, both resulting functions are coherently combined (18). The symbol “circshift” denotes the circular shift of the signal samples vector.
(18)$$\begin{array}{l} R_{mSBZP\_SC}\left[\tau ,f_{d},n\right]=R\left[\tau ,f_{d},n\right]+\underset{-N_{Z}}{\mathrm{circshift}}\left(R\left[\tau ,f_{d},n+1\right]\right)=\\ \sum_{i=0}^{N-1}\left(\begin{array}{l} \;\;\;\;\;\;\;\;\;\;\;\;\;\;SC[\left(i-n\right)\mathrm{mod}N]\mathrm{IFFT}\left\{\mathrm{FFT}\left\{r[t]g\left(t,i\right)e^{-j2\pi f_{d}t}\right\}\circ \mathrm{FFT}{\left\{c[t]\right\}}^{*}\right\}+\\ \underset{-N_{Z}}{\mathrm{circshift}}\left(SC[\left(i-\left(n+1\right)\right)\mathrm{mod}N]\mathrm{IFFT}\left\{\mathrm{FFT}\left\{r[t]g\left(t,i\right)e^{-j2\pi f_{d}t}\right\}\circ \mathrm{FFT}{\left\{c[t]\right\}}^{*}\right\}\right) \end{array}\right) \end{array}$$

The algorithm is illustrated in Figure 17 using the *W*’-function to derive its loss. The received signal *r*[*t*] within blocks of the primary code length *T* with SC has an example *τ* = (1/2)*T*. Then, the signal is block-by-block correlated using two SC replicas for *SC*[*n*] and *SC*[*n* + 1]. Thus, two *W*’-functions called the *W*’_1_ and the *W*’_2_ are formed according to cross-correlated parts of the signal and each replica. Then, due to the proposed coherent combining (18), the original loss (17) is equal to the mean value of the sum of both *W*’-functions (19) for the mSBZP case. Two partial correlations form one full correlation. The zero result of (19) indicates that the loss has been removed, and the main peak is not split.
(19)$$Loss_{mSBZP\_SC}\left[\tau \right]=-20\log_{10}\left(1-\frac{1}{T}\sum_{t=0}^{\left(N-1\right)T}\left[W{’}_{1}\left[t,\tau \right]+W{’}_{2}\left[t,\tau \right]\right]\right)\;=-20\log_{10}\left(1-\frac{1}{T}\left(\sum_{t=0}^{\tau -1}1+\sum_{t=\tau }^{T-1}1\right)\right)\;=0\;\;;\tau \in [0,T)$$

Equation (18) could be written in the post-correlation form (20) and the corresponding schema in Figure 18.
(20)$$\begin{array}{l} R{\left(\tau ,f_{d},n\right)}_{mSBZP\_SC}=\sum_{i=0}^{N-1}SC[i-n]\gamma +\underset{-N_{Z}}{\mathrm{circshift}}\left(\sum_{i=0}^{N-1}SC[i-\left(n+1\right)]\gamma \right)\\ \mathrm{where}:\gamma =\mathrm{IFFT}\left\{\mathrm{FFT}\left\{r[t]g\left(t,i\right)e^{-j2\pi f_{d}t}\right\}\circ \mathrm{FFT}{\left\{c[t]\right\}}^{*}\right\} \end{array}$$

### 6.2. The mPCA-Based Joint Primary and SC Estimator (mPCA mSBZP for SC Algorithm)

Finally, Equation (18) could be realized in the form of the proposed estimator (21) by the help of the mPCA algorithm, using the mSBZP and the property of linearity. The algorithm was realized by combining two mPCA units working with different consecutive *n* and *n* + 1. It is illustrated in the algorithm, block-computing schema in Figure 19. The “D” symbol means a circular shift back by *N_z_* samples. The results of both schemas are the same and follow. However, the post-correlation approach in Figure 18 requires *N* computations of the PCS algorithm per SC period *NT*, whilst its mPCA version requires only one. It is an essential computational complexity benefit.
(21)$$\begin{array}{l} R{\left(\tau ,f_{d},n\right)}_{mPCA\_mSBZP\_SC}=\mathrm{IFFT}\left\{\mathrm{FFT}\left\{\sum_{i=0}^{N-1}\left[\lambda \right]e^{-j2\pi f_{d}t}\right\}\circ \mathrm{FFT}{\left\{c[t]\right\}}^{*}\right\}\\ \mathrm{where}:\lambda =SC[i-n]r[t]g\left(t,i\right)+\underset{-N_{Z}}{\mathrm{circshift}}\left(SC[i-\left(n+1\right)]r[t]g\left(t,i\right)\right) \end{array}$$

### 6.3. The mPCA-Based Joint Primary and SC Estimator (mPCA mSBZP for SC Algorithm)—Results

The results of the proposed mPCA mSBZP schema from Figure 19 are presented in Figure 20. These results were compared with the previous conventional post-correlation DBZP results [21,23] from Figure 6. The results in an entire interval of the Galileo E1C CS25_1_ code period were depicted in Figure 21 and Figure 22. The resulting functions of the *PNR* and the *FSPR* were constant for a proper *n*. Thus, they were independent of the *τ* as was intended for the joint estimator. The proposed algorithm surmounts the post-correlation DBZP in both the *PNR* and the *FSPR* ratios. A better *PNR*, and a main to SC side peaks ratio, respectively, was also evident here. 

The overall performance of the proposed algorithm was compared with other algorithm estimators, using the Monte-Carlo simulation (for [*τ*,*f_d_*,*n*] set of tested parameters and 3000 simulation points) and the Constant-False-Alarm-Rate (CFAR, comparing *PNR* with fixed threshold) detection in Figure 23 at the final. The detection results of both above-compared schemas (post-DBZP and mPCA mSBZP) were similar. However, the mPCA algorithm surmounts others in lower computational complexity (number of PCS operations per SC period *NT*). See Section 6.4.

Figure 24 presents an example of the resulting CAF. The proposed mPCA mSBZP algorithm is unbiased in the *f_d_* and the *τ* estimation. The remaining fragments are observable, but it has no significant effect. Its origin is in a non-zero mean value of SC.

### 6.4. Computational Complexity and Implementation Remarks

The proposed PCS mPCA mSBZP algorithm (Figure 19) obtains a beneficial reduction of computational complexity. It is based on a realization of the PIT in the pre-correlation, the pre-PCS stage using the mPCA, unlike its achievement in the post-correlation, the-post PCS stage (Figure 18). The PCS algorithm using demanding FFT resources is used only once per an entire SC period *NT*, unlike *N*-times use. Above that, the computational complexity is reduced by a factor of two by utilization of the SBZP, unlike the commonly used DBZP. Thus, twice as smaller *N_FFT_* is required for the same resolution of the computed cross-correlation function.

Then, resources (time, latency, FPGA chip area, or power consumption) can be laid out more economically. The PCS unit and its FFT blocks can be designed with a considerably longer latency equal to *NT*, unlike *T*, or with a lower clock frequency (lower power consumptions) or with shared FFT resources (smaller chip area).

The proposed algorithm was presented here on simulations using the Galileo E1C 25 bits long code CS25_1_. However, the algorithm was approved using the Galileo CS4_1_ (Galileo E5b-I) and the BeiDou B1 20 bits long Neumann–Hoffman SC as well. The algorithm was designed for an existing PCS based acquisition unit with the PCS core, which is implemented in System-On-a Chip (SoC) FPGA with three 16K (16,384) FFT blocks. This unit will be usable for the acquisition of these advanced signals using this easily implementable mPCA mSBZP principle in the future.

This algorithm is generally applicable for the acquisition of tiered GNSS signals with short to medium-long SC (length < 100, see Table 1), acquisition in weak signal conditions. Its utilization is limited by its serial search in the SC code phase and the number of Doppler frequency shift bins due to a long PIT. Nevertheless, the PIT of all similar methods using the PCS algorithm is also limited in the same way due to the effect of non-zero *f_d_* on the code rate.

## 7. Conclusions

The article offers an analysis and a comparison of the PCS algorithm based acquisition methods of GNSS tiered signals with SC. Then, it proposes a new approach to joint primary and SC phase estimator. It focuses on a technique using the SBZP and the SC pre-correlation coherent accumulation processing.

The SBZP offers a better efficiency of available FFT resources in comparison with the DBZP. Nevertheless, it is affected by a formation of parasitic fragments in the CAF in conjunction with signals containing a navigation data or SC bit transition. Thus, its results depend on a primary code phase. Therefore, the straightforward SBZP is not directly suitable for the implementation of the intended joint estimator.

The article offers an analysis of these adverse effects and corresponding loss. Then, the proposed mPCA mSBZP algorithm for SC is designed, using an effective combining of cross-correlation results for neighboring SC phases. The mSBZP allows use of the SBZP in the presence of SC bit transition without parasitic fragments in the CAF. The mPCA processing is used for a required coherent time extension over SC bits. The algorithm uses the PCS algorithm for a primary code search, while the SC is searched sequentially. The sequential search is slower than the parallel one, but the SC is considerably shorter than the primary. Nevertheless, it also allows acquisition of signals combining SC with a navigation message data.

The proposed algorithm was verified and compared with other similar acquisition algorithms to confirm its comparable functionality using the PNR, the FSPR, and the detection probability. 

Besides the suppression of the aforementioned adverse effects and the more efficient utilization of the *N_FFT_* than the DBZP, the mPCA mSBZP has other beneficial properties. The algorithm is *τ*, and *f_d_* unbiased. The essential benefit is a decrease in the number of PCS operations that are required per a period of a tiered signal from *N* to one PCS operation (where *N* is the number of SC bits). It allows savings of PCS computational resources (hardware resources or latency).

## Figures and Tables

**Figure 1 sensors-20-02965-f001:**
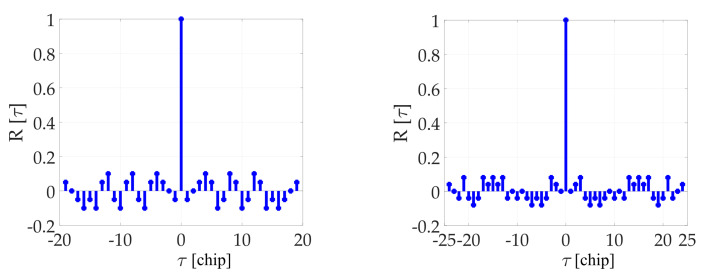
BeiDou-B1 (Neumann–Hoffman, **left**) and Galileo E1C (CS25_1_, **right**) autocorrelation function of SC.

**Figure 2 sensors-20-02965-f002:**
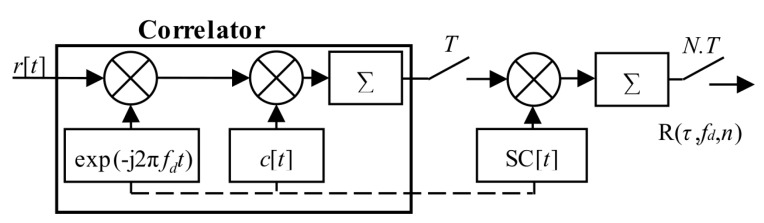
Sequential search acquisition schema.

**Figure 3 sensors-20-02965-f003:**
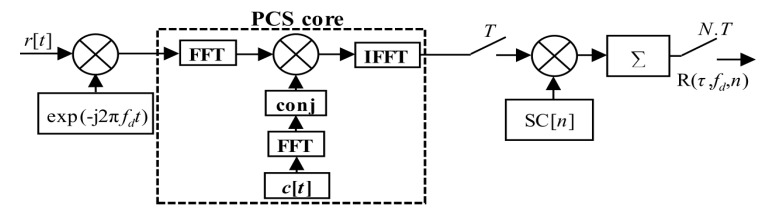
Acquisition schema using PCS, the straightforward single-step post-correlation approach.

**Figure 4 sensors-20-02965-f004:**
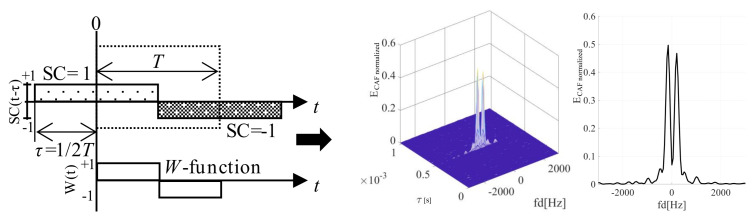
*W*-function (**left**), the effect of SC bit transition on the CAF (**middle**) and Doppler shift domain (**right**), for *τ* = (1/2)*T*.

**Figure 5 sensors-20-02965-f005:**
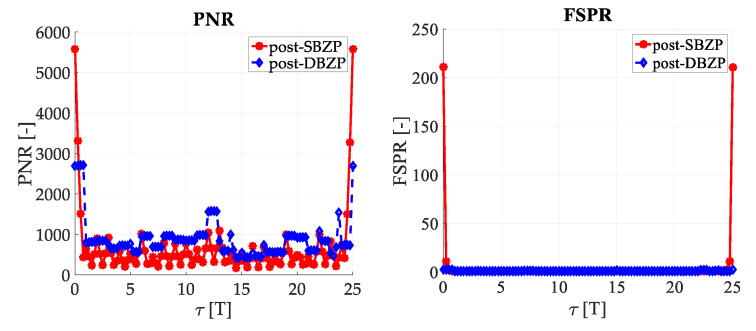
Result of the utilization of the single-step PCS post-correlation schema with SC (Galileo CS25_1_).

**Figure 6 sensors-20-02965-f006:**
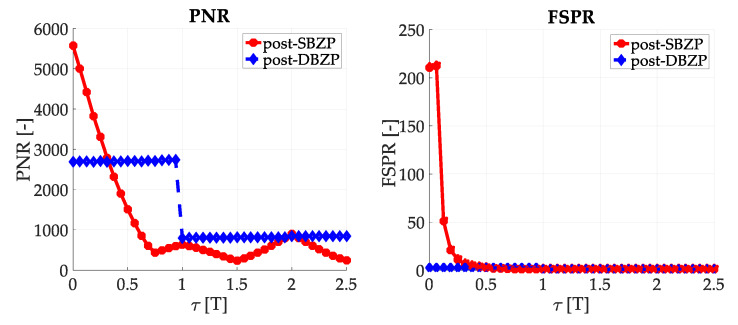
Result of the utilization of the single-step PCS post-correlation schema with SC, detail of Figure 5.

**Figure 7 sensors-20-02965-f007:**
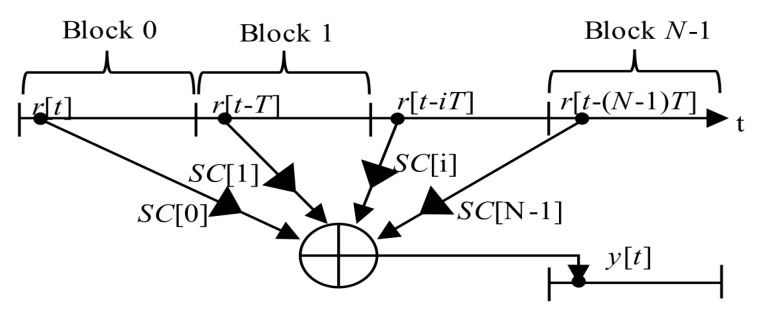
Functional diagram of mPCA.

**Figure 8 sensors-20-02965-f008:**
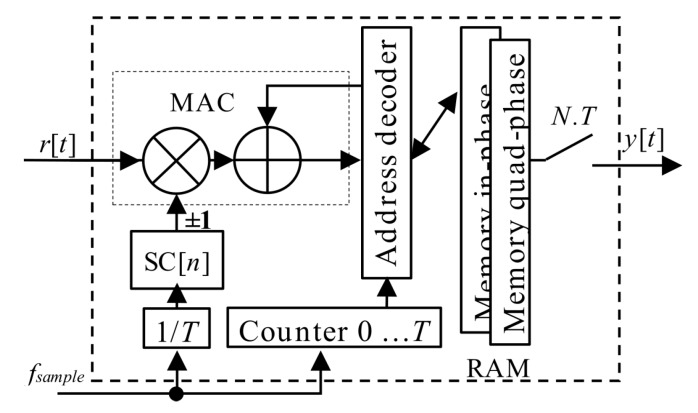
Block diagram of mPCA unit with MAC.

**Figure 9 sensors-20-02965-f009:**
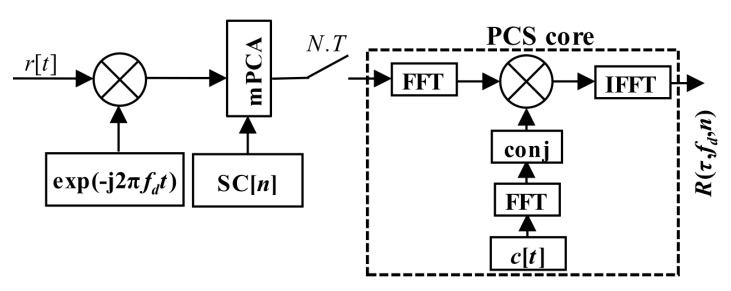
PCS and mPCA pre-correlation processing of SC.

**Figure 10 sensors-20-02965-f010:**
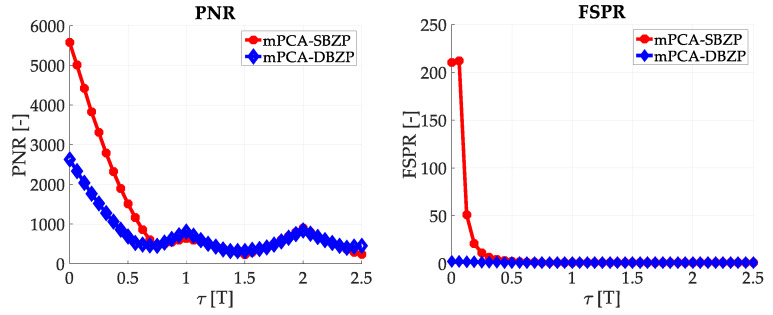
Result of PCS, mPCA pre-correlation processing of SC.

**Figure 11 sensors-20-02965-f011:**
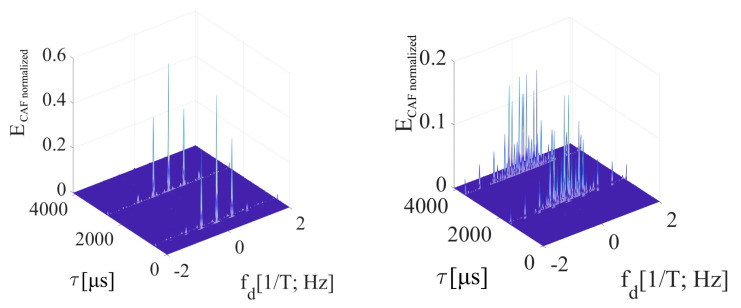
Result of mPCA pre-correlation processing of SC using DBZP, *τ* = (1/4)*T* (**left**) and *τ* = (3/4)*T* (**right**).

**Figure 12 sensors-20-02965-f012:**
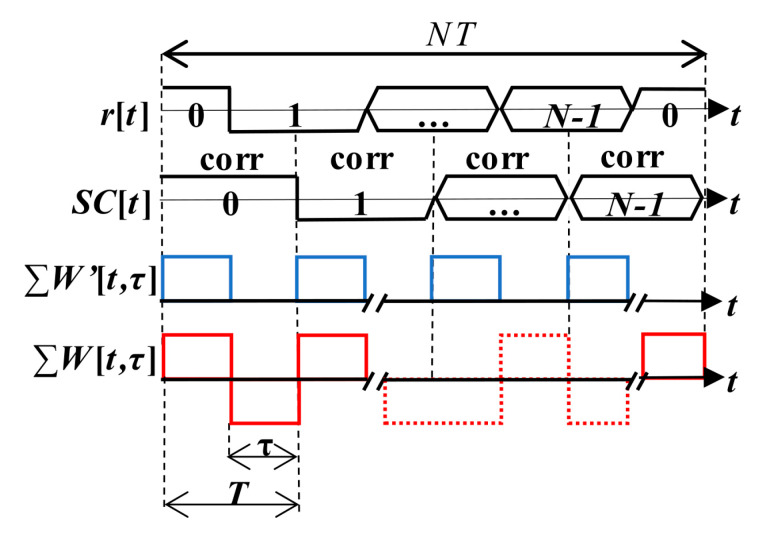
Description of the *W* and *W*’-function for *τ* = (1/2)*T*.

**Figure 13 sensors-20-02965-f013:**
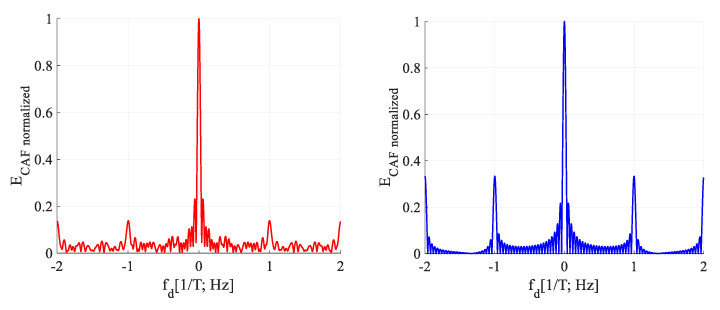
Splitting of the false peak (**left**) and the main peak (**right**) in the Doppler shift domain for *τ* = (1/4)*T*.

**Figure 14 sensors-20-02965-f014:**
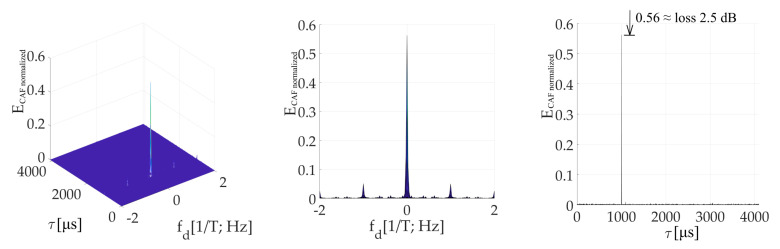
Resulting CAF of the mPCA pre-correlation processing of SC using the SBZP, for *τ* = (1/4)*T*. CAF (**left**), Dopler shift domain (**middle**), Code phase domain (**right**).

**Figure 15 sensors-20-02965-f015:**
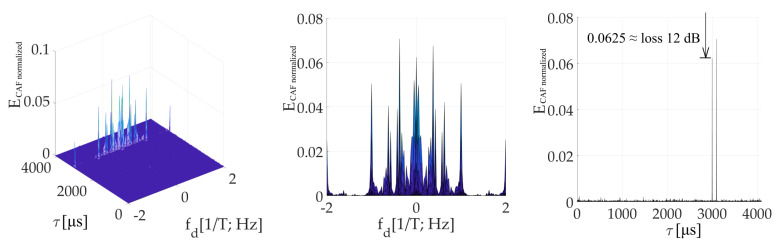
Resulting CAF of the mPCA pre-correlation processing of SC using the SBZP, for *τ* = (3/4)*T*. CAF (**left**), Dopler shift domain (**middle**), Code phase domain (**right**).

**Figure 16 sensors-20-02965-f016:**
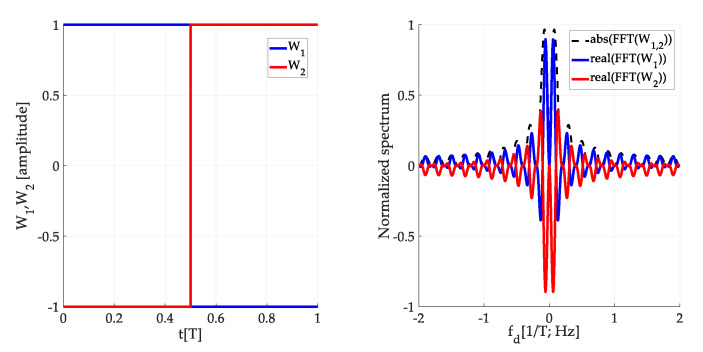
Description of *W_1_* and *W_2_*-function in the time domain (**left**) and the spectral domain (**right**) for *τ* = (1/2)*T*.

**Figure 17 sensors-20-02965-f017:**
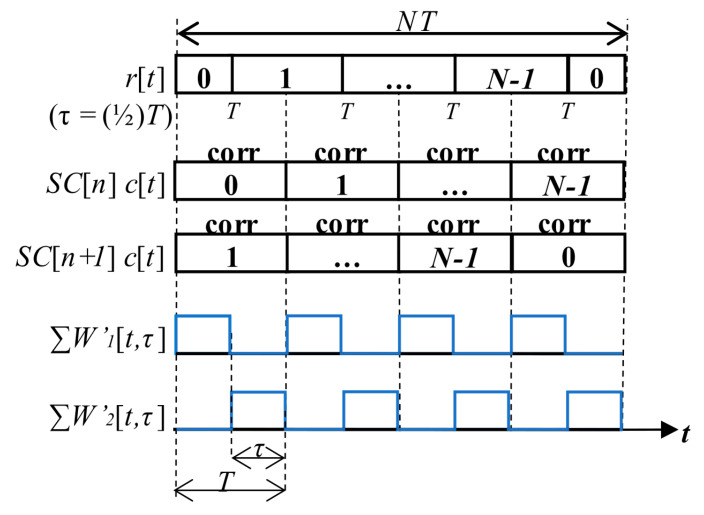
The mSBZP for SC algorithm, *W*’_1_ and *W*’_2_ -functions.

**Figure 18 sensors-20-02965-f018:**
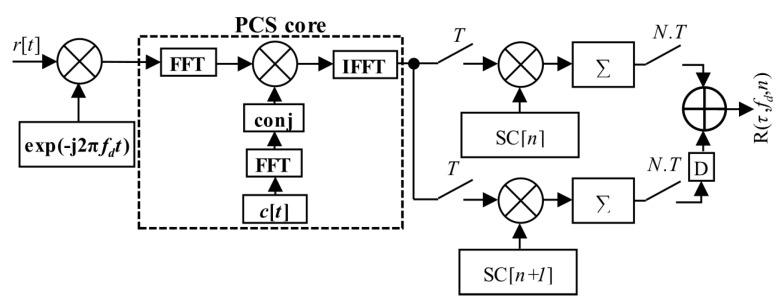
Proposed mSBZP schema for SC acquisition in post-correlation processing.

**Figure 19 sensors-20-02965-f019:**
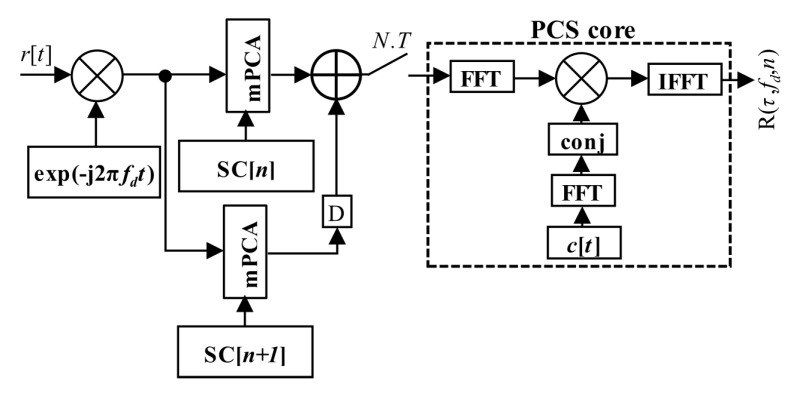
Proposed mPCA mSBZP schema for SC acquisition in the pre-correlation processing algorithm.

**Figure 20 sensors-20-02965-f020:**
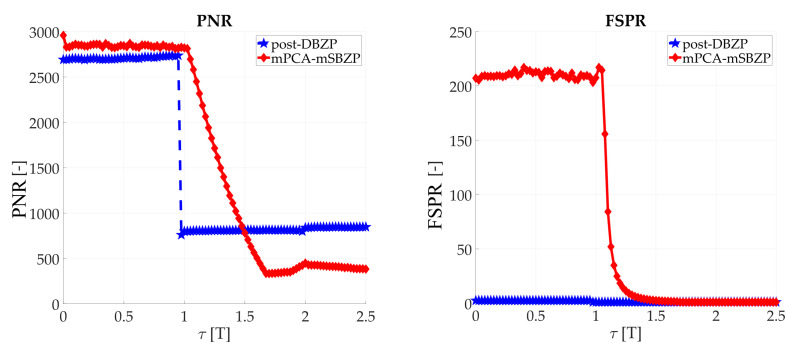
Results of the proposed modified processing, in first bits of SC.

**Figure 21 sensors-20-02965-f021:**
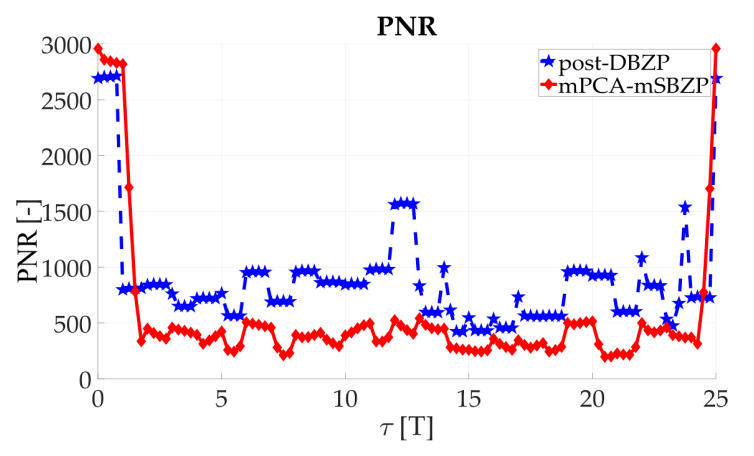
PNR result of the proposed modified processing, in an entire period of SC.

**Figure 22 sensors-20-02965-f022:**
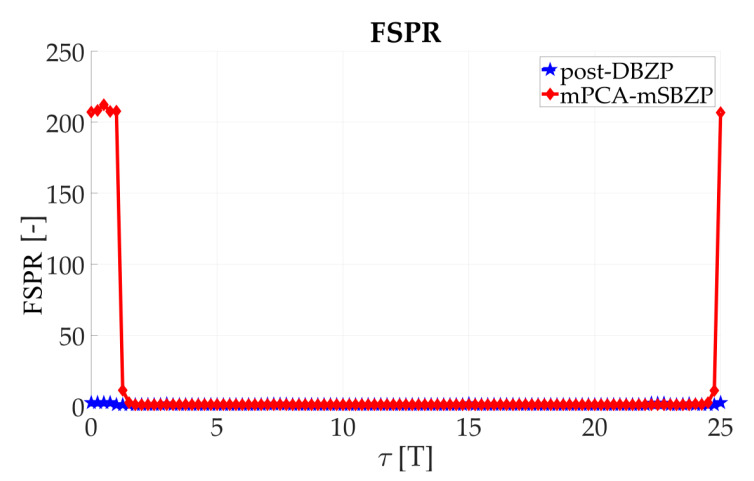
FSPR result of the proposed modified processing, in an entire period of SC.

**Figure 23 sensors-20-02965-f023:**
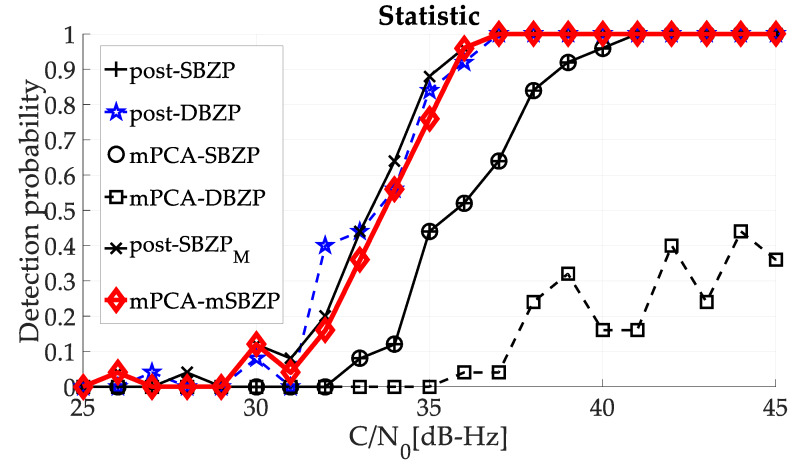
Detection probability performance of mPCA mSBZP algorithm for SC acquisition.

**Figure 24 sensors-20-02965-f024:**
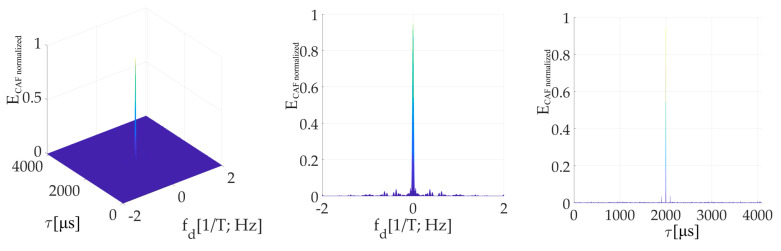
CAF of mPCA mSBZP for SC algorithm for *τ* = (1/2)*T*. CAF (**left**), Dopler shift domain (**middle**), Code phase domain (**right**).

**Table 1 sensors-20-02965-t001:** Examples of Global Navigation Satellite Systems (GNSS) signals and secondary code (SC).

Signal	Signal Properties				
	Primary Code *N* [chip]/*T* [ms]		Secondary Code *N* [chip]/*T* [ms]		Data [bit/s]
**GPS L1C-P**	10,230	10	1800	18,000	-
**GPS L5-I**	10,230	1	10	10	50
**GPS L5-Q**	10,230	1	20	20	-
**Galileo E1C**	4092	4	25	100	-
**Galileo E5a-I**	10,230	1	20	20	25
**Galileo E5a-Q**	10,230	1	100	100	-
**Galileo E5b-I**	10,230	1	4	4	125
**Galileo E1b-Q**	10,230	1	100	100	-
**BeiDou B1-I**	2046	1	20	20	50

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
