# Peer review of "Joint Acquisition Estimator of Modern GNSS Tiered Signals Using Block Pre-Correlation Processing of Secondary Code"

_sensors, 2020, doi:10.3390/s20102965_

Round 1
Reviewer 1 Report
=== Major Comments ===
Authors proposed a new approach to joint primary and secondary code phase estimator based on the Single Block Zero-Padding (SBZP) and the secondary code pre-correlation coherent accumulation processing (PCA). The proposed algorithm was compared and verified using the Peak to Noise Ratio, the First to Second Peak Ratio, and the detection probability with others to confirm its excellent properties.
However, there are many parts of mistakes in writing that must be improved.
=== Minor Comments ===
p.1, line 29-31:
References [1,2,4] are cited in the text ambiguously.
- [1] is about tiered code, but it is cited in the text to assist the description on general aspect of GNSS.
- [2] is about remote Earth observation, but it is cited to describe time synchronization. So, a profer citation on time synchronization is needed.
- Misra is used for the reference on ‘indoor positioning’ issue, but Misra does not touch this issue. I recommend to use profer references.
p.2, line 48:
- Text in line 48-49 is ambiguous to understand. Please clarify the sentence to convey your meaning.
p.2, line 51:
- Table 1 should be updated to include the signals having secondary code (e.g. Galileo E6-C, BeiDou B1C, B3I, B2I, B2a)
- data [bit/s] or data [simbol/s] ?
p.2, line 52, 56:
Technical differences between linear autocorrelation and cyclic autocorrelation should be mentioned before these figures.
p.3, line 59-60:
References [6-8] are not need here (or in this paper).
p.3, line 63:
In order to clarify the meaning of the sentence, I propose to include the following text: Commonly used approaches (for the tiered signal acquisition) are based on a two-step schema.
p.4, line 141:
r[t] in (2) represents the received signal that is a function of tau, n, f_d as seen in (3). The term after r[t] in (2) seems to be terms for local replica. However, these are not well matched to Fig. 3. (c[t] or c[t-tau]?, SC[n] or SC[t-n]?)
D_nav in (2) is for local replica? Probabely, this should be removed.
p.5, line 147:
Subsection 3.2 comes without 3.2.
p.5, line 157:
(6), (7) at the end of sentence should be removed?
p.5, line 158:
The Single Block Zero-Padding (SBZP) (6) causes ~
-> The Single Block Zero-Padding (SBZP) written as (6) causes ~ ....??
p.5, line 165:
The Double Block Zero-Padding (DBZP) (7) causes ~
-> The Double Block Zero-Padding (DBZP) written as (7) causes ~ ....??
p.6, line 198:
in Figure 4 in Section 3.3 -> ~ Section 3.2 ....?
p.7, line 209:
NFFT (the length of FFT) is determined by sampling frequency and the duration of primary code duration. The duration of primary code of Galileo E1C is 4ms. So, for Nz=96, the sampling frequency of 1 MHz seems to be used. Please add more detailed information on this issue.
p.8, line 232:
Text in (15) is broken.
p.8, line 237:
In Fig. 8, r[t] -> r[t’] ??
p.8, line 239:
Check acronyms for Multiply-And-Accumulate (MAC).
p.9, line 264:
In Fig. 12, the unit of tau is missing. The same thing happens in Fig. 25.
p.12, line 334:
Eq. (18) should be explained in more details.
p.13, line 367:
Text in line 367 is not well matched to (20). More detailed explanation should be added.
p.13, line 369:
What do a) and d) in Fig. 18 mean? Also, all figures in this paper are poor to describe the technical meaning. This will polute author’s claim.
p.14, line 398:
What is y-label in Fig. 21? This also happens in Figs. 22 and 23.
p.15, line 407:
In the text, authors described that the peak energy is dissipated by false peak. This may be interpreted that the first peak of FSPE has relatively low energy that will affect detection probability seriously. However, the difference between FSPRs of mPCA-mSBZP and post-DBZP is seen very clearly in Fig. 23 but there is no difference in their detection probabilities. The authors’ claim on the performance improvement in weak signal detection by the use of the proposed method should be explained in more details.
Overall,
- Citation of references should be reviewed.
- Uses of abbreviation before abbreviation definition in text are seen many time in the text.
- Abbreviations and originals are mixed used frequently.
- There are many sentences with overlapping contents
Reviewer 2 Report
Dear Editor,
I send to you my suggestions for this paper:
- All acronyms from the paper must be explained.
- In Abstract, please add the information about obtained results from research test.
- The quality of Figure 1,2,5,8,9,10,12,15,16,17,19,20,25 should be better.
- The symbols in all Equations mus be explained in text, e.g. Equations (1,2,3,4,5 etc.).
- Please check if the units on the axis on the selected Figure are described, see Figure 15,16,25
- The conclusion must included information why proposed method is better than another solution. The obtained results from test must be described in conclusion.
- The proofreader should also check the English language of paper.
Round 2
Reviewer 1 Report
== Minor comments ==
p.6 line 195:
‘~ provided in Figure 7.’should be ‘~ provided in Figure 6.’?
p.8 line 229:
- As similar to Figure 8, y[t`] in Figure 7 should be y[t]?
- ‘*+-1’ in Figure 8 is probably related to the dual ported memory. More detailed explanation for this should be added.
p.15 line 386:
It should be ‘...with the Constant-False-Alarm-Rate (CFAR).....’
CFAR comes first here in this paper without any description. More detailed description on this should be added.
